# Using MicroRNA Arrays as a Tool to Evaluate COVID-19 Vaccine Efficacy

**DOI:** 10.3390/vaccines10101681

**Published:** 2022-10-08

**Authors:** Yen-Pin Lin, Yi-Shan Hsieh, Mei-Hsiu Cheng, Ching-Fen Shen, Ching-Ju Shen, Chao-Min Cheng

**Affiliations:** 1Institute of Biomedical Engineering, National Tsing Hua University, Hsinchu 300, Taiwan; 2Taiwan Business Development Department, Inti Taiwan, Inc., Hsinchu 302, Taiwan; 3Department of Pediatrics, National Cheng Kung University Hospital, College of Medicine, National Cheng Kung University, Tainan 701, Taiwan; 4Department of Obstetrics and Gynecology, Kaohsiung Medical University Hospital, Kaohsiung Medical University, Kaohsiung 807, Taiwan

**Keywords:** microRNA array, COVID-19, mRNA vaccine, pregnancy, immune response

## Abstract

In order to solve COVID-19 pandemic, the entire world has invested considerable manpower to develop various new vaccines to temporarily alleviate the disaster caused by the epidemic. In addition to the development of vaccines, we need to also develop effective assessment methods to confirm vaccines’ efficacy and maximize the benefits that vaccines can bring. In addition to common evaluation methods, vaccine-specific and temporal expression of microRNAs have been shown to be related to vaccine efficacy or vaccine-associated diseases. In this article, we have introduced a microRNA-array-based approach, which could be potentially used for evaluating COVID-19 vaccine efficacy, specifically for pregnant women. As the mRNA in mRNA vaccines is decomposed by host cells within a few days, it is considered more suitable for pregnant women to utilize the method of vaccination during pregnancy. Moreover, pregnant women belong to a high-risk group for COVID-19, and there is currently no appropriate vaccine to newborns. Therefore, it’s important to find improved tools for evaluation of vaccine efficacy in response to the current situation caused by COVID-19.

## 1. Introduction

COVID-19 is a global infectious disease caused by SARS-CoV-2 which starting from a cluster of pneumonia cases in Wuhan, Hubei Province, China at the end of 2019 [1]; the cause was identified as a SARS-CoV-2 and quickly spread to many surrounding countries in early 2020, evolving into a global pandemic. In order to improve this situation, the world has invested considerable manpower and resources to develop various COVID-19 vaccines. Vaccines developed in response to the outbreak have had an effective and positive impact; however, efficacy assessment methods should be used to confirm their efficacy and maximize vaccine benefits.

The mRNA vaccine does not contain any virus—it contains the genetic code (mRNA) of the spike protein on the surface of the SARS-CoV-2 virus. This is a new technology [2] to stimulate the body’s own immune response. These vaccines contain messages from mRNA, usually constructed from foreign proteins produced by pathogens (such as viruses) or cancer cells as blueprints [3]. These messages allow the body to produce this antigen on its own and the cells in our body then present the antigen on their surfaces, triggering the desired specific immune response. Henceforth, if the body is exposed to a virus, the immune system already recognizes those specific antigens from the vaccine and can fight the infection quicker and in a targeted manner. As the mRNA in mRNA vaccines is decomposed by host cells within a few days, it is considered more suitable for pregnant women to use this vaccination method during pregnancy. In addition, pregnant women and newborns are a high-risk group, meaning more effective vaccine assessment methods are needed to ensure their health.

Vaccine-specific and temporal expression of microRNAs have been shown to be related to vaccine efficacy or vaccine-associated diseases. Atherton et al. (2019) found microRNA patterns specific to vaccine types, and saw microRNAs as potential biomarkers that could provide valuable insights for vaccine development [4]. Oshiumi H. (2021) suggested the importance of extracellular vesicle microRNAs as tools to improve vaccine efficacy and to act as biomarkers in predicting immune response and adverse reactions after vaccinations [5]. Small regulatory microRNAs also have fundamental roles in regulating the expression and functions of key immunological mediators such as cytokines [6,7,8]. These research publications have found microRNAs to be involved in many immune regulatory pathways and have potential applications in vaccine research. 

## 2. Methods

Vaccination is the most commonly used method in the world today to prevent the spread of bacteria and viruses. Vaccination against COVID-19 can not only prevent infection, but it can also protect us from serious illness or death from COVID-19. However, assessing vaccine efficacy is an important step toward vaccine selection. Comparing and evaluating the effectiveness of each vaccine can provide better vaccination recommendations and facilitate significant follow-up impacts.

There are primarily two forms of vaccine efficacy evaluation methods: (1) humoral immunity [9], referred to as “antibody production” [10] (Antibody production); and (2) cellular immunity, which can be roughly divided into T cell response [11] and QuantiFERON Array [12,13]. In addition to the above methods, we hope that we can also evaluate vaccine efficacy via a microRNA expression profiling array (Figure 1). We intend to assess the effects of COVID-19 vaccination among vaccinated pregnant women and non-vaccinated pregnant women by investigating real-time microRNA expression profiles with a MIRAscan and NextAmp™ Analysis System. 

The NextAmp™ Analysis System was developed as a molecular diagnostic device designed to detect and analyze the gene expression of multiple biomarkers based on polymerase chain reaction amplification technology. The core component of the system facilitating multi-gene analysis is a 36 mm × 36 mm × 1 mm reaction chip called a PanelChip^®^, which consists of 2500 nanowells, with each nanowell representing one real-time PCR reaction well [10]. MIRAscan is a microRNA PanelChip^®^ consisting of 83 different microRNAs related to various diseases. MIRAscan microRNA analysis service is provided by Inti Taiwan, Inc., whose vision is to help increase IVF success rates through more personalized and accessible molecular testing solutions. These microRNA candidates were selected from miDatabase™, a comprehensive microRNA database consisting of data from over 30,000 publications. After a sample was loaded into the MIRAscan microRNA PanelChip^®^, it was then loaded into Q Station™ for qPCR reaction and subsequent analysis, resulting in raw Cq data values depicting microRNA expression levels.

The resulting microRNA expression profiles were normalized, and microRNAs without amplification signals across all profiles were removed. Based on the experimental design, the number of differentially expressed microRNAs for each comparison were identified (|ΔCq| ≥ 1). Once the differentially expressed microRNAs were found, miRTarBase was used for microRNA target interaction (MTI) analysis. miRTarBase is one of the largest databases of experimentally validated microRNA-target interactions (Figure 2). We filtered out MTIs with less than 3 reference support and non-functional MTIs. Gene set enrichment analysis using clusterProfiler was then performed on the resulting gene list from MTI. Gene ontology, KEGG pathway and disease ontology were used for functional analysis due to their long-standing curation. We hope the resulting microRNA data can be used for vaccine efficacy assessment and to provide a reference for subsequent vaccine administration planning.

## 3. Results—Clinical Samples from Two Pregnant Women

The entry of SARS-CoV-2 into human host cells is mediated by the SARS-CoV-2 spike protein located on the surface of the virus [14]. An mRNA vaccine for COVID-19 provides our bodies with the code to produce the non-infectious viral spike protein in order to direct cells to help stimulate a natural immune response. This response is mainly achieved through the production of T cells and neutralizing antibodies against SARS-CoV-2, which circulate in the body and immediately bind to the virus and prevent it from entering cells, thus protecting us from getting sick easily. T cells help the immune system fight intracellular infections and can also kill infected cells directly. Thus, in contrast to traditional vaccines, mRNA vaccines do not contain any viral proteins themselves, but only the information our own cells need to produce the viral signature that triggers the desired immune response [15]. Each of the three COVID-19 vaccines described above induces an immune response against SARS-CoV-2, and after our first encounter with a particular bacterium or virus, in the years or decades that follow, adaptation cells can remember them -this is what we call immune memory [16], if you come into contact with a real virus or bacteria in the future, the immune system will remember it, produce antibodies against it, and quickly activate the right immune cells, thereby killing viruses or bacteria and protecting us from disease.

In this study, maternal blood samples from pregnant women were collected after the doctor personally explained the research study content and obtained the patients agreement for participating in the study. Patients then signed the consent form for specimen collection that also included information such as vaccine type, dose, gestational weeks at the time of administration, side effects, etc. for subsequent analysis. Maternal blood samples were subsequently collected during delivery time. These samples were from the patients at Taiwan’s Kaohsiung Medical University Chung-Ho Memorial Hospital (KMUHIRB-SV(II)-20210087).

The microRNAs detected in the plasma of pregnant women who had received three doses of the Moderna vaccine (M1) and pregnant women who had not received any vaccine (M2) were analyzed, and the ΔCq (∆Cq(M1 vs. M2)) of each microRNA was calculated. To identify differentially expressed microRNAs, the following selection criteria was applied: |ΔCq| ≥ 1 (including ΔCq ≥ 1 and ΔCq ≤ −1). Comparative analysis showed that 7 microRNAs had |ΔCq| values greater than 1 between sample source types: hsa-miR-1972, hsa-miR-191-5p, hsa-miR-423-5p (∆Cq (M1-M2) < −1); hsa-miR-16-5p, hsa-miR-486-5p, hsa-miR-21-5p, hsa-miR-451a (∆Cq (M1-M2) > 1) (Table 1). When comparing ∆Cq (M1-M2) values, those that were negative indicated that microRNAs were overexpressed in samples from subjects that received three doses of COVID-19 vaccine (M1) compared to samples from subjects that received no vaccine (M2). 

Gene set enrichment analysis of pathway terms and gene ontology (GO) terms were performed using the differentially expressed microRNAs as input. The background gene set based on validated microRNA target interaction were from miRTarbase. A total of 3 of the pathways identified in the top 10 biological processes Enrichment GO terms are closely related to immune regulatory pathways after vaccination, including positive regulation of protein modification process, positive regulation of tumor necrosis factor superfamily cytokine production, and adaptive immune response (Table 2). The top 10 biological process pathway Enrichment GO terms are organized in a network, where each pathway is a node and edges represent gene overlap between pathways. Mutually overlapping gene sets tend to cluster together, making it easy to quickly identify the major enriched functional themes and interpret the enrichment results (Figure 3).

## 4. Discussion

Most microRNAs regulate gene expression by inhibiting protein translation or by degrading the mRNA transcript. A single microRNA may regulate the expression of multiple genes and its encoded proteins. microRNAs are not only involved in regulating the innate immune system, but also have been implicated in regulating adaptive immunity by controlling the development and activation of T and B cells [17]. During the past few years, many microRNAs have been found to be important in the development, differentiation, survival, and function of B and T lymphocytes, dendritic cells, macrophages, and other immune cell types. After vaccination, innate sensors are triggered by the intrinsic adjuvant activity of the vaccines, resulting in production of type I interferon and multiple pro-inflammatory cytokines and chemokines. RNA sensors such as Toll-like receptor 7 (TLR7) and MDA5 are triggered by the mRNA vaccines [18]. Researchers demonstrated that IFN-g expressions in NK cells after 1st vaccine doses correlated with SARS-CoV-2 vaccine-induced neutralizing antibody [19]. Analyzing the expression of immune related proteins and cytokines has the potential to be a tool for assessing the relevance of vaccine-induced immunity. Vaccine efficacy depends on immune responses, such as proinflammatory cytokine production and lymphocyte activation. Proinflammatory cytokine production are caused by immune responses to antigens, leading to production of antigen-specific antibodies. The immune-regulatory microRNA levels in serum extracellular vesicles (EVs), such as miR-148a levels were associated with specific antibody titers, and could be potential biomarkers for vaccine efficacy [20]. 

Preliminary small-scale experiments were carried out on a new PCR array-based platform for samples that were collected from vaccinated and unvaccinated pregnant women. After analyzing the microRNA Cq values from each data set, seven microRNAs with different expression between vaccinated and unvaccinated pregnant women samples were found. Among these seven microRNAs, hsa-miR-486-5p was also found to be differentially expressed in plasma between pregnant women in their first trimester compared to non-pregnant women [21]. MicroRNAs have been found to regulate different mechanisms specific to pregnant women as substantial changes occur in the body to support the developing fetus [21]. Further experiments are still needed for the differential expression of microRNAs of participants with physiological status in the future to find out whether pregnant women have unique microRNA expression profiles due to specific immune regulations.

According to the GO databases, among the identified genes regulated by differentially expressed microRNAs, 14 genes (such as interleukin-6 (IL6), signal transducer and activator of transcription 3 (STAT3), Toll-like receptor 3 (TLR3), etc.) were involved in the positive regulation of the tumor necrosis factor superfamily cytokine production pathway, while 24 genes participated in the adaptive immune response pathway (Table 3). Both of these biological pathways are related to immune regulation. Through their effect on the production of cytokines and proteins related to immune regulation, microRNAs may further affect the production of antibodies. More experiments are still needed to confirm the relationship between the differential expression of microRNAs, the production of cytokines and proteins, and antibody response.

## 5. Conclusions

The roles and regulatory mechanisms of these microRNAs in immune regulation still require additional examination to confirm the relationship between microRNAs and antibody responses among different physiological status and backgrounds. However, through this preliminary study, the production of antibodies after vaccination can be linked to the regulation of genes and microRNAs before the protein translation process. For subsequent vaccine efficacy evaluation, microRNA expression may be used as a valuable tool for real-time monitoring of antibody effects.

## Figures and Tables

**Figure 1 vaccines-10-01681-f001:**
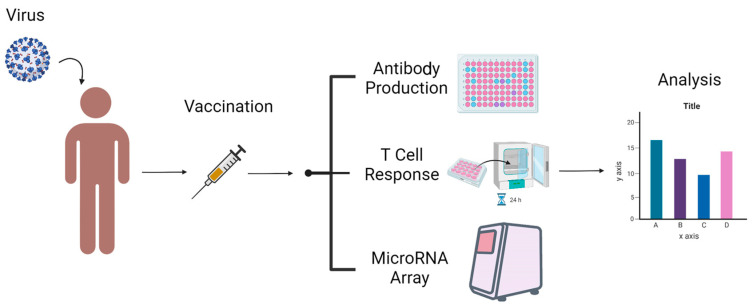
Three methods to evaluate COVID-19 vaccines effectiveness.

**Figure 2 vaccines-10-01681-f002:**
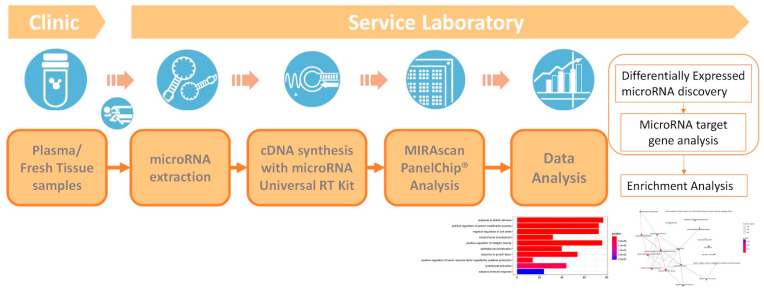
microRNA candidate discovery services provided by Inti Taiwan Inc. with MIRAscan microRNA assay. MIRAscan is a microRNA PanelChip^®^ consisting of 83 different microRNAs related to various diseases. There are two spike-in controls for the MIRAscan microRNA assay: one is RT spike-in control, the other is qPCR spike-in control. These two controls are used to ensure consistency and quality for the cDNA synthesis process and qPCR reaction process, respectively.

**Figure 3 vaccines-10-01681-f003:**
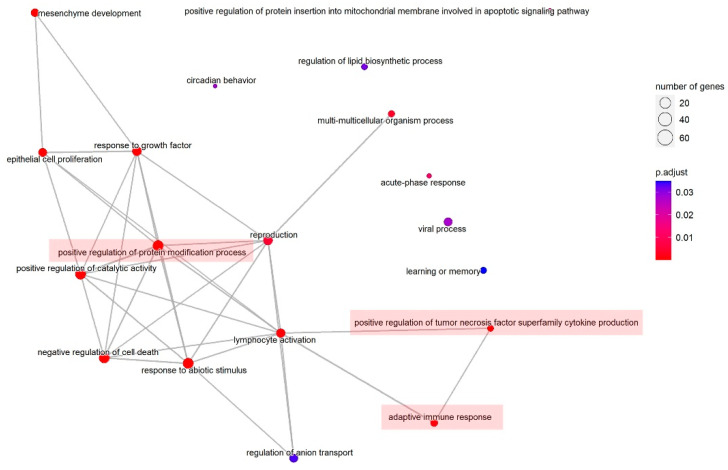
Enrichment map of the top 10 biological process (BP) Enrichment GO terms organizes them into a network with edges connecting overlapping gene sets. The input genes used for enrichment analysis are regulated by the differentially expressed microRNAs identified between the plasma samples of two pregnant women, one who received three doses of COVID-19 vaccine while the other received no vaccinations. Pathways highlighted in red indicate close relationships to immune regulatory pathways after vaccination.

**Table 1 vaccines-10-01681-t001:** Clinical information and differential microRNA expression between two pregnant women with different vaccination histories.

Clinical Data			
	**3 doses (M1)**		**No dose (M2)**
**Parity**	**2**		**3**
**Age (year)**	**36**		**40**
**BMI**	**21.797**		**28.377**
**Weeks of gestation at delivery**	**40**		**39**
**Kind of COVID-19 vaccine for first/ second/ third dose**	**Moderna/Moderna/Moderna**		**-/-/-**
	**Normalized Cq value**		
	**3 doses (M1)**	**No dose (M2)**	**ΔCq (M1-M2)**
**hsa-miR-1972**	7.8294	11.0089	−3.1795
**hsa-miR-191-5p**	9.2544	11.2722	−2.0178
**hsa-miR-423-5p**	10.7744	12.7122	−1.9378
**hsa-miR-16-5p**	8.3869	6.1593	2.2276
**hsa-miR-486-5p**	9.8724	7.5511	2.3213
**hsa-miR-21-5p**	9.1844	7.6378	1.5466
**hsa-miR-451a**	8.0427	5.0866	2.9561

MIRAscan was used to detect 83 microRNAs’ expression profiles in plasma samples collected from two pregnant women. Additionally, selection criteria to identify differentially expressed microRNAs were as follows: |ΔCq| ≥ 1 (including ΔCq ≥ 1 and ΔCq ≤ −1); microRNAs highlighted in orange indicate overexpression of microRNAs in the vaccinated, three-dose group (M1) while those highlighted in green represent overexpression of microRNAs in the unvaccinated, no-dose group (M2).

**Table 2 vaccines-10-01681-t002:** Top 10 biological process Enrichment Gene Ontology (GO) terms.

ID	Description	GeneRatio	BgRatio	*p* Value (Adjust)
GO:0009628	response to abiotic stimulus	77/232	441/2689	1.06 × 10^−7^
GO:0031401	positive regulation of protein modification process	73/232	433/2689	1.28 × 10^−6^
GO:0060548	negative regulation of cell death	73/232	439/2689	1.67 × 10^−6^
GO:0060485	mesenchyme development	32/232	130/2689	9.37 × 10^−6^
GO:0043085	positive regulation of catalytic activity	76/232	494/2689	1.07 × 10^−5^
GO:0050673	epithelial cell proliferation	40/232	205/2689	4.97 × 10^−5^
GO:0070848	response to growth factor	54/232	333/2689	0.00014
GO:1903557	positive regulation of tumor necrosis factor superfamily cytokine production	14/232	39/2689	0.00022
GO:0046649	lymphocyte activation	44/232	265/2689	0.00050
GO:0002250	adaptive immune response	24/232	115/2689	0.00109

GeneRatio = Ratio of input genes that were annotated per term. Input genes are regulated by the differentially expressed microRNAs analyzed from the Cq value data sets. BgRatio = ratio of all genes that were annotated in this term. The *p*-values were adjusted using the “BH” (Benjamini-Hochberg) method.

**Table 3 vaccines-10-01681-t003:** Differentially expressed microRNAs and their target interaction genes participated in the two pathways related to immune regulation in the top 10 enrichment Gene Ontology (GO) terms of biological process.

ID	Pathway Description	** Gene ID	* Differentially Expressed microRNAs
GO:1903557	positive regulation of tumor necrosis factor superfamily cytokine production	APP/CLU/HMGB1/IFNG/IL1A/ IL6/IL12B/MIF/MYD88/PIK3R1/ STAT3/TLR3/BCL10/RASGRP1	hsa-miR-16-5p hsa-miR-21-5p hsa-miR-191-5p hsa-miR-451a hsa-miR-486-5p
GO:0002250	adaptive immune response	JAG1/BCL6/CLU/MTOR/MSH6/ HMGB1/ICAM1/IFNG/IL1B/IL6/ IL6R/IL12A/IL12B/SMAD7/MEF2C/ MSH2/MYD88/STAT3/TAP1/TSC1/ UNG/BCL10/DUSP10/ICOSLG	hsa-miR-16-5p hsa-miR-21-5p hsa-miR-451a

* The differentially expressed microRNAs selected were identified between the plasma samples of two pregnant women, one who received three doses of COVID-19 vaccine while the other received no vaccinations. ** Gene ID= gene-centered information at NCBI website. APP: amyloid beta precursor protein, CLU: clusterin, HMGB1: high mobility group box 1, IFNG: interferon gamma, IL1A: interleukin 1 alpha, IL6: interleukin 6, IL12B: interleukin 12B, MIF: macrophage migration inhibitory factor, MYD88: MYD88 innate immune signal transduction adaptor, PIK3R1: phosphoinositide-3-kinase regulatory subunit 1, STAT3: signal transducer and activator of transcription 3, TLR3: Toll-like receptor 3, BCL10: BCL10 immune signaling adaptor, RASGRP1: RAS guanyl releasing protein 1, JAG1: jagged canonical Notch ligand 1, BCL6: BCL6 transcription repressor, MTOR: mechanistic target of rapamycin kinase, MSH6: mutS homolog 6, ICAM1: intercellular adhesion molecule 1, IL1B: interleukin 1 beta, IL6R: interleukin 6 receptor, IL12A: interleukin 12A, SMAD7: SMAD family member 7, MEF2C: myocyte enhancer factor 2C, MSH2: mutS homolog 2, TAP1: transporter 1, ATP binding cassette subfamily B member, TSC1: TSC complex subunit 1, UNG: uracil DNA glycosylase, DUSP10: dual specificity phosphatase 10, ICOSLG: inducible T cell costimulator ligand.

## Data Availability

Not applicable.

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
