# Peer review of "Using MicroRNA Arrays as a Tool to Evaluate COVID-19 Vaccine Efficacy"

_vaccines, 2022, doi:10.3390/vaccines10101681_

Round 1
Reviewer 1 Report
Dear Authors, your comunication is pretty interisting, but I have some comments for you
First of all in your introduction paragraph, in my opinion you should "announce" your study so it is better that you mention that the samples of the study come from pregnant women and what mRNA vaccine you are considering.
Methods- It is good to specify the number of pregnant women you enrolled in your study, this could give more static relevance to your data.
Did you get blood sample from the same preganent woman before the vaccination?
Line 67-75 Describe a short list of the Gens and pathways identified and add a table list with those gene would, it would complete the meaning of what you are trying to comunicate
Line 108-114- In line 108 you talk about 7 micRNAs with ∆Cq values grater than 1, but you describe after looks incongruent or at least not too clear.
Reviewer 2 Report
The manuscript is interesting but extremely short. The abstract has three lines, the introduction is very limited as well as the scope of the manuscript and the discussion. Data validation is also an issue since there is no information about the individuals analyzed fertility patients and then pregnant women. Are they different? It is confusing. The screening and chip involve more miRNA than those in the table, in fact, figure 2 refers to the protocol, one part is impossible to visualize it.
Table 1 should include more information on pregnant or non-pregnant. What are the controls of those assays. There should be information on the targets.
Table 2 are you sure is abiotic stimulus?
Figure 3 seems to represent the responses of pregnant women please correct the legend.
There is no discussion and the conclusion needs editing
Round 2
Reviewer 1 Report
The manuscript can be accept in the present form
Author Response
Thank you so much!
Reviewer 2 Report
The manuscript was improved from the previous one. There is. However, some information missing concerning the age and conditions of pregnant women. Table one it states weeks of gestation, and there is no information. it is especially important in the two groups due to the type of miRNA analyzed. Table 3 was improved. I strongly recommend the authors to separate results from the discussion since it tends to confuse the reader, specially since there is a lack of discussion of the critical miRNA found as compared to the literature in non pregnant women. The conclusions should also be modified.
